# Volatile Dimethyl Disulfide from Guava Plants Regulate Developmental Performance of Asian Citrus Psyllid through Activation of Defense Responses in Neighboring Orange Plants

**DOI:** 10.3390/ijms231810271

**Published:** 2022-09-07

**Authors:** Siquan Ling, Hualong Qiu, Jinzhu Xu, Yanping Gu, Jinxin Yu, Wei Wang, Jiali Liu, Xinnian Zeng

**Affiliations:** 1Guangdong Engineering Research Center for Insect Behavior Regulation, College of Plant Protection, South China Agricultural University, Guangzhou 510642, China; 2Guangdong Provincial Key Laboratory of Silviculture, Protection and Utilization, Guangdong Academy of Forestry, Guangzhou 510520, China; 3Plant Protection Research Institute, Guangdong Academy of Agricultural Sciences, Guangzhou 510640, China

**Keywords:** sulfur volatiles, dimethyl disulfide, eavesdropping, defense response, guava, sweet orange, Asian citrus psyllid

## Abstract

Intercropping with guava (*Psidium guajava* L.) can assist with the management of Asian citrus psyllid (ACP, *Diaphorina citri* Kuwayama), the insect vector of the huanglongbing pathogen, in citrus orchards. Sulfur volatiles have a repellent activity and physiological effects, as well as being important components of guava volatiles. In this study, we tested whether the sulfur volatiles emitted by guava plants play a role in plant–plant communications and trigger anti-herbivore activities against ACP in sweet orange plants (*Citrus sinensis* L. Osbeck). Real-time determination using a proton-transfer-reaction mass spectrometer (PTR-MS) showed that guava plants continuously release methanethiol, dimethyl sulfide (DMS), and dimethyl disulfide (DMDS), and the contents increased rapidly after mechanical damage. The exposure of orange plants to DMDS resulted in the suppression of the developmental performance of ACP. The differential elevation of salicylic acid (SA) levels; the expression of phenylalanine ammonia lyase (PAL), salicylate-O-methyl transferase (SMT), and pathogenesis-related (PR1) genes; the activities of defense-related enzymes PAL, polyphenol oxidase (PPO), and peroxidase (POD); and the total polyphenol content were observed in DMDS-exposed orange plants. The emission of volatiles including myrcene, nonanal, decanal, and methyl salicylate (MeSA) was increased. In addition, phenylpropanoid and flavonoid biosynthesis, and aromatic amino acid (such as phenylalanine, tyrosine, and tryptophan) metabolic pathways were induced. Altogether, our results indicated that DMDS from guava plants can activate defense responses in eavesdropping orange plants and boost their herbivore resistance to ACP, which suggests the possibility of using DMDS as a novel approach for the management of ACP in citrus orchards.

## 1. Introduction

Plants are susceptible to attack by herbivores and pathogens during their life cycle, and they have evolved a variety of strategies to effectively defend themselves against invaders, including pre-existing physical barriers and inducible defense responses [1,2]. Volatile organic compounds (VOCs) are considered to be very effective chemical signals in plant–plant–insect communication. In preparation for possible upcoming threats, undamaged plants can continuously monitor, detect, perceive, and respond to surrounding volatile signals with great sensitivity and discrimination [3,4]. Usually, herbivore- and pathogen-attacked plants release increase their amount of volatiles, including green leaf volatiles (GLVs), terpenoids, and other aromatic compounds. These volatiles can have systematic effects on neighboring plants or within plants, and are similar to phytohormones, acting as direct elicitors, precursors of defense compounds, or as volatile signals to induce or prime plant defense responses [5,6,7,8]. Meanwhile, some plant species can also emit high levels of flavor and aroma compounds, even when they are not damaged or stressed [9]. These constitutively emitted volatiles have significant implications for the receiving plants and insects. For example, volatile-mediated chemical communications of weed–barley [10], onion–potato [11,12], mint–soybean [9], molasses grass–maize [13], and guava–citrus [14] resulted in receiver plants that were less attractive and unsuitable for herbivores and more attractive to parasitoids. The ability of receiver plants to respond to specific pant volatiles can significantly minimize fitness costs and avoid unnecessary and costly responses [15,16]. Thus, volatile-induced changes in the allocation of defense compounds may affect plant resistance to biotic stresses, which ultimately affects the persistence of plants in the community [17].

Guava (*Psidium guajava* L.) is widely planted in many tropical and subtropical countries. Guava volatiles have attracted intensive interest since it was reported that guava intercropped with citrus groves had a lower incidence of huanglongbing (HLB) disease and a reduced infestation of Asian citrus psyllid (ACP, *Diaphorina citri* Kuwayama) [18,19,20]. ACP is an important sap-sucking insect pest of citrus because it transmits phloem-limited bacteria (*Candidatus* Liberibacter spp.), which is strongly implicated in HLB disease [21], one of the most destructive citrus diseases and having a devastating impact on the citrus industry [22,23]. Studies have shown that sesquiterpene hydrocarbons are the predominant volatile terpenoids of guava leaves, and (*E*)-β-caryophyllene is the major volatile, independently of the cultivar and the location [14,24,25,26]; although, the emission rate of (*E*)-β-caryophyllene from guava plants changed under different developmental stages and at different time points [27]. Furthermore, sulfur volatiles, including hydrogen sulfide (H_2_S), sulfur dioxide (SO_2_), methanethiol, dimethyl sulfide (DMS), dimethyl disulfide (DMDS), methional, and dimethyl trisulfide (DMTS), are also released by guava [28,29]. It has been demonstrated that certain volatiles, especially sesquiterpene (*E*)-β-caryophyllene and sulfur volatile DMDS, could play an important role in ACP reduction by functioning as repellents or by interfering against it, thereby limiting the pathogen spread [26,27,28,30,31]. Meanwhile, our previous studies revealed a new mechanism by which citrus exposure to guava volatiles alters the citrus olfactory cues, thus reducing the attractiveness of citrus to ACP [32]. Further laboratory work showed that citrus plants can eavesdrop on guava volatiles such as (*E*)-β-caryophyllene and (*E*)-4,8-dimethyl-1,3,7-nonatriene (DMNT), to prime their anti-herbivore activities [14].

In addition to aroma, sulfur volatiles are also very effective in terms of physiological effects. For example, exposure to H_2_S and DMS can increase sulfur-induced resistance (SIR) in crops [33]. H_2_S may be involved in plant defense signaling by regulating glutathione metabolism, inducing the expression of pathogenesis-related genes and other defense-related genes, regulating enzyme activity through post-translational modifications, and interacting with phytohormones [34]. DMDS exposure significantly upregulates the expression of defense-related genes in tomato; in particular, the upregulation of PR1 and PR5 suggests that the salicylic acid (SA) pathway is involved in the induction of systemic resistance to fungi [35]. Furthermore, DMTS, one of the main activity components identified in Chinese leek (*Allium tuberosum*) volatiles, can induce the expression of the defense-related genes of apple fruit and suppress ring rot disease [36]. Therefore, certain sulfur volatiles can act as volatile signals, to induce defense responses in neighboring plants. In the context of guava–citrus interactions, however, it is unclear whether and how guava-released sulfur volatiles promote defense responses in citrus plants.

This study was aimed at determining the effect of sulfur volatiles released from guava on the defense responses of sweet orange (*Citrus sinensis* L. Osbeck) plants, as well as explaining the corresponding mechanisms of induced resistance against ACP. First, methanethiol, DMS, and DMDS from guava plants were determined in real time, using a proton-transfer-reaction mass spectrometer (PTR-MS). On this basis, the effects of the exogenous application of sulfur volatile(s), especially DMDS, on the levels of herbivore resistance, defense-related gene expression, phytohormones, enzyme activities, and volatile and non-volatile metabolites of orange plants were studied. The results will further explain the ecological and metabolomic mechanisms of guava–citrus–psyllid interactions, and reveal the possibility of using DMDS as a novel approach for the management of ACP in citrus orchards.

## 2. Results

### 2.1. Dimethyl Disulfide Is the Dominant Headspace Sulfur Volatile of Guava Plants

The results of the chemical analyses on the sulfur volatiles associated with headspace from guava plants are shown in Figure 1. Several sulfur volatiles, including methanethiol (*m/z* 49), DMS (*m/z* 63), and DMDS (*m/z* 95), were positively identified and monitored in real time by PTR-MS, with high overall emission rates. In the undamaged condition, the concentrations of methanethiol, DMS, and DMDS were 0.19, 0.93, and 1.83 ppbv, respectively. After mechanical damage, the concentrations of the three sulfur volatiles in headspace increased immediately, and were 0.34, 3.47, and 16.28 ppbv, respectively, after 15 min of treatment, which were significantly different to the undamaged plants. Meanwhile, the concentrations of sulfur volatiles gradually decreased and returned to the normal level (6 h for DMDS). Overall, it was found that DMDS is the major headspace sulfur volatile in guava plants.

### 2.2. The Developmental Performance of Psyllids Was Suppressed on Dimethyl Disulfide-Exposed Orange Plants

In wind tunnel tests, no significant difference was observed in the settlement of ACP on orange plants after exposure to sulfur volatiles compared with the control (Figure 2a). Meanwhile, significantly more ACP adults selected methyl salicylate (MeSA)-exposed orange plants than the control. In oviposition cages, the number of eggs oviposited on orange plants that had been exposed to sulfur volatiles and MeSA was not significantly different compared with the control (Figure 2b). In addition, the exposure of orange plants to methanethiol and DMS at the current concentrations did not affect nymphal development. However, conditions were unsuitable for hatched nymphs to grow on DMDS-exposed, 10 × DMDS-exposed (equaled to DMDS released by mechanically damaged guava), or MeSA-exposed orange plants (Figure 2c). The development of nymphs was delayed, with a significant increase in the percentage of 1st–3rd instar nymphs and a significant decrease in the percentage of 5th instar nymphs. Both the number of emerged adults and the survivorship from egg to adult were considerably reduced (Figure 2d). These data imply that the orange plants became more resistant to ACP after exposure to DMDS released at concentrations from damaged and undamaged guava.

### 2.3. The Content of Salicylic Acid and the Expression of Defense-Associated Genes Were Induced in Dimethyl Disulfide-Exposed Orange Plants

To evaluate the effects of sulfur volatiles on the defense responses of neighboring orange plants, we measured the levels of jasmonic acid (JA) and SA in orange leaves that had been exposed to methanethiol, DMS, and DMDS for 14 d, and to MeSA for 3 d. The exposed orange plants showed no significant differences in JA levels compared with the control (Figure 3a). The levels of SA in the orange plants that had been exposed to methanethiol and DMS were not significantly different from the control (Figure 3b). The plants exposed to DMDS and 10 × DMDS showed an increase in SA levels, and this was 1.49- and 2.11-fold higher than that in control leaves, respectively. Similarly, the SA levels of the plants exposed to MeSA also increased significantly.

Next, the transcript levels of SA-related genes including phenylalanine ammonia lyase (PAL), salicylate-O-methyl transferase (SMT), and pathogenesis-related (PR1) protein in exposed orange leaves were quantified. None of these genes showed enhanced expression after exposure to the used concentrations of methanethiol and DMS (Figure 4a,b). After exposure to DMDS, the expression of PAL was not significantly different compared with the control, while the expression of SMT (7 d) and PR1 (3 d), in contrast, was significantly induced (Figure 4c). In addition, exposure to 10 × DMDS induced the expression of PAL, SMT, and PR1, with effects similar to those of the MeSA treatment (Figure 4d). Taken together, these results imply that exposure to DMDS and 10 × DMDS causes an overall induction of SA-dependent defense responses in orange plants. For subsequent experiments, the orange plants that had been exposed to DMDS and 10 × DMDS for 14 d were used for biochemical parameter determination and metabolomics analysis.

### 2.4. The Activity of Defense-Associated Enzymes and the Content of Total Polyphenol Were Enhanced in Dimethyl Disulfide-Exposed Orange Plants

The PAL activity in the DMDS, 10 × DMDS, and MeSA treatments of orange leaves was 1.09-, 1.69-, and 2.04-fold higher than those in the control, respectively (Figure 5a). Consistent with this result, polyphenol oxidase (PPO) activity increased in the exposed plants, and was 1.25-, 1.47-, and 1.58-fold higher than those in control leaves, respectively (Figure 5b). Peroxidase (POD) activity in the orange leaves treated with DMDS and 10 × DMDS was induced, but the difference was not significant compared with the control (Figure 5c). Similarly, the total polyphenol content was increased (1.44- and 1.54-fold, respectively) in the DMDS- and 10 × DMDS-treated leaves, but was not significantly different from those in the control (Figure 5d). Exposure to MeSA significantly increased the total polyphenol content of the orange leaves. The results showed that exposure to DMDS increased the activity of defense-associated enzymes and the total polyphenol content; however, its effect was limited at concentrations equivalent to intact guava emissions.

### 2.5. Volatile and Semipolar Metabolomics of Orange Plants Were Changed after Exposure to Dimethyl Disulfide

Gas chromatography-mass spectrometry (GC-MS) analysis of the volatiles collected from exposed and control orange plants revealed differences in the volatile profile. The first two components of a partial least squares-discriminant analysis (PLS-DA) explained 60.7% of the total variation (Figure 6a). A total of 27 major compounds were consistently released by the orange plants, based on different treatments (Figure 6b; Appendix A). Within 22 of them were terpenoids; including monoterpene pinene, myrcene, limonene, β-ocimene, and linalool; sesquiterpene β-elemene, (*E*)-β-caryophyllene, and α-farnesene; and homoterpene DMNT and TMTT. After treatment with DMDS, the release of MeSA was significantly increased, but the total amount of volatiles did not change significantly. Meanwhile, after exposure to 10 × DMDS, the total volatiles increased, of which nonanal, decanal, and MeSA were significantly increased, and limonene and myrcene also had varying degrees of elevation. In addition, MeSA treatments significantly affected the profile of volatile metabolites in the neighboring orange plants; in particular, monoterpenes (such as sabinene, α/β-pinene, myrcene, limonene, β-ocimene, and linalool), aldehydes (octanal, nonanal, and decanal), and MeSA were significantly increased (Figure 6c).

A total of 671 non-volatile metabolites were isolated and identified by ultra-high-performance liquid chromatography tandem mass spectrometry (UHPLC-MS/MS) combined with databases. PLS-DA analysis with all metabolites explained 61.9% of the data set variation in the first two components (Figure 7a). The metabolite features showed a clear separation between the different groups, suggesting that there were large metabolic differences between the plant response to the sulfurs and MeSA compared with the control (Figure 7b). After treatment with DMDS and 10 × DMDS, a total of 31 and 155 metabolites were upregulated, and 21 and 128 metabolites were downregulated, respectively (Figure 7c). Similarly, MeSA treatment caused a large number of changes in metabolite features. The differentially expressed metabolites (DEMs) included benzenes, carboxylic acids (mainly amino acids), cinnamic acids, coumarins, fatty acyls, flavonoids, indoles, keto acids, organooxygen compounds, organonitrogen compounds, phenols, prenol lipids, and steroids and their derivatives (Appendix A). Venn diagram analysis revealed that the 10 × DMDS- and MeSA-treated DEMs were similar, with 105 upregulated and 90 downregulated metabolites being identical (Figure 7d). Furthermore, 11 upregulated and 13 downregulated metabolites were the same in the three comparison groups, respectively. KEGG pathway analysis revealed that the DEMs in the “DMDS vs. Control” group were significantly enriched in tyrosine metabolism and tryptophan metabolism pathways (Figure 7e). In addition to amino acid (tyrosine, lysine, alanine, aspartate, glutamate, phenylalanine, arginine, proline, cysteine, and methionine) metabolism pathways, the 10 × DMDS and MeSA treatments were also involved in defense-related secondary metabolite biosynthesis pathways, such as phenylpropanoid and flavonoid biosynthesis. Moreover, the glutathione metabolism pathway and arginine biosynthesis were significantly enriched after treatment with 10 × DMDS and MeSA, respectively.

## 3. Discussion

Volatiles of plant origin have been reported to play a vital role in regulating plant defenses against herbivores. In response to these volatile signals, a plant can exhibit a multitude of adaptation responses, including changes in their signal transduction, transcriptome, proteome, and metabolome. In guava–citrus intercropping systems, high emission levels of guava volatiles play an important role in guava–citrus–psyllid interactions [14,26,32]. A previous study detected and identified seven sulfur volatiles from guava leaves [28]. Similarly, volatile methanethiol, DMS, and DMDS were also positively identified and monitored in real time using PTR-MS in the present study. Among them, DMDS had the highest content and became the dominant headspace sulfur volatile after wounding. On this basis, we investigated the effects of sulfur volatile exposure on ACP performance. The results showed that the development of ACP was considerably negatively affected and the number of emerged adults was reduced after exposure to DMDS, especially at the concentration released by guava after damage (10 × DMDS). Volatiles can be adsorbed and re-released in plant leaf surface waxes (passive plant–plant interactions), thereby interfering with herbivorous insects [37]. However, a series of defense-related genes and metabolites were induced in the orange plants after exposure to DMDS. Similarly to the induction of systemic resistance to pathogens in tomato [35], DMDS was found to be an important volatile component that mediates active plant–plant interactions between guava and orange plants, rather than passive plant–plant interactions.

Volatile cues can trigger different plant signaling cascades, which can interact with each other positively or negatively [15]. The early events are usually associated with Ca^2+^ signaling, reactive oxygen species (ROS) accumulation, mitogen-activated protein kinase (MAPK) cascades, and phytohormone signaling [2,38], leading to the activation of downstream targets, to elicit the biosynthesis of defense-related signal molecules [39]. For example, GLVs, indole, (*E*)-nerolidol, and α-farnesene trigger JA biosynthesis and signaling in plants [2,6,40,41]. In contrast, the monoterpenes pinene and camphene trigger SA-associated immunity [42]. JA and SA can act as secondary messengers in several physiological processes of plants, as well as in defense responses. In fact, JA and SA pathways interact through a complex network of regulatory interactions, including priming, synergistic effects, and mutual antagonism [43]. It has been reported that the JA pathway mainly induces resistance against chewing herbivores and necrotrophic pathogens, whereas the SA pathway mainly confers resistance against biotrophic pathogens and most sucking/piercing insects [2,44]. Our analysis revealed that orange plants exposed to DMDS showed an increase in SA levels, especially after treatment with 10 × DMDS, and the result was similar for MeSA exposure. Furthermore, the upregulation of SA-regulated genes, including PAL, SMT, and PR1, was consistent with the changes in SA level. Recent studies have shown that H_2_S and DMDS induce SA-associated systemic resistance to fungi in plants [34,35]. Therefore, the induction of herbivore resistance to ACP should be consistent with the activation of SA-associated defense responses in orange plants.

The reprogramming of metabolic pathways eventually triggers the biosynthesis and secretion of defense-related metabolites. Plants produce toxic metabolites such as phytoalexins and protease inhibitors, and secondary metabolites such as phenols and flavonoids, which can directly affect the feeding, growth, and reproduction of herbivores [14,45]. It is well known that PPO and POD are important plant oxidative enzymes that act against herbivores by reducing their food intake and nutrient digestibility [46,47]. PAL is a key enzyme for the synthesis of phenolic compounds (such as flavonoids and phenylpropanoids), and it plays a crucial role in the defense against sucking insects [48]. Thus, the increased accumulation of PPO, POD, and PAL activities and total phenol content investigated in this study may partly explain the higher direct defense against ACP in the DMDS- and MeSA-exposed orange plants. Moreover, plants also produce volatiles to negatively affect herbivorous behavior and/or to indirectly protect plants, by attracting the natural enemies of herbivores. The findings of the present research showed that exposure to DMDS induced the release of monoterpenes such as myrcene, aldehydes such as nonanal and decanal, as well as MeSA. It has been found that adults of *Tamarixia radiata*, a specialist parasitoids of ACP nymphs, are attracted to MeSA [49]. However, MeSA and myrcene have also been found to have attraction effects on ACP adults [14,50], which may reduce the interference effect of guava volatiles on ACP host selection.

In the current study, tyrosine, tryptophan, phenylalanine, and other amino acid metabolic pathways were significantly changed after DMDS (or 10 × DMDS) or MeSA exposure. Furthermore, the defense pathways regulated by classical phytohormones and distinct amino acid metabolic pathways, which are pivotal for the biosynthesis of many protective plant natural products, constitute integral parts of plant defenses [51]. Aromatic amino acids such as phenylalanine, tyrosine, and tryptophan are important molecules in plant metabolism, and are the precursors for the natural plant products that play critical roles in plant growth, defense, development, environmental factors, and reproduction [52]. For example, phenylalanine and tyrosine act as exogenous precursors of wheat secondary metabolism through PAL-associated pathways [53]. Therefore, the changes in the amino acid metabolic pathways in the present study may also have contributed to the resistance of the exposed plants to ACP.

Notably, the guava natural release concentration of DMDS induced a weak defense effect in orange plants, while treatment with the damage release concentration significantly induced defense responses. It has been reported that volatile doses and exposure time impact perception in neighboring plants [54]. In addition, plants modulate the strength of their responses, based on the reliability of the detected volatile cues, while the integration of multiple volatile cues is often more reliable and robust than a single volatile signal [55,56]. Previously, we identified that guava volatile terpenoids such as (*E*)-β-caryophyllene and DMNT can prime the JA-related anti-herbivore activities of orange plants [14]. Methanethiol and DMS at the current concentrations have limited effect on the defense response of orange plants, although it has been found that DMS can increase the resistance of crops [33]. Nevertheless, in guava–citrus interactions, DMDS should be part of a more complex volatile blend, forming signals with other compounds, even when released from undamaged guava plants. Combined with the crosstalk of plant internal signaling, such as phytohormone JA and SA, the defense responses are ultimately stimulated, thus enhancing resistance to ACP.

Meanwhile, plant defense is costly, as the activation of defense responses requires a significant investment of resources toward defense production [57,58]. These costs are typically measured as a reduction in plant fitness (e.g., survivorship, growth, and reproduction) in the absence of herbivores, and thus tradeoffs exist between investment in defenses and investment in growth and reproduction [59,60,61]. Interestingly, the perception of early reliable cues may allow a plant to anticipate a probable attack and prime its defenses before herbivory occurs. This priming leads to more rapid and/or stronger defense responses after a real attack, which is a cost-saving strategy that restricts the deployment of expensive and specialized defensive metabolites until necessary [62,63]. In the current study, DMDS released from undamaged guava induced a low-level expression of defense-related genes and metabolites in orange plants, although the developmental performance of ACP nymphs was significantly reduced. We hypothesize that DMDS at the current concentration primes defense responses in orange plants. In addition, there are tradeoffs between defenses against herbivores and pathogens, or even against different herbivore species, so that defense against one may increase susceptibility to another, which would be consistent with an SA–JA tradeoff [60,64,65]. Thus, the utility of DMDS in field applications will depend on the impact at the community level. It will be important to evaluate the ecological costs and fitness benefits of the DMDS-induced defense response in citrus plants, as it could be beneficial for farmers, in terms of a citrus pest management plan.

## 4. Materials and Methods

### 4.1. Plants and Insect Rearing

Guava (*P. guajava* L. cv. Yanzhi) and sweet orange (*Citrus sinensis* Osbeck cv. Hongjiang) plants (1.5 years old, 40–50 cm tall), obtained from the College of Horticulture, South China Agricultural University (SCAU), Guangzhou, China, were grown in pots (10 cm diameter, 16 cm height) with loamy soil. The potted orange and guava plants were separately maintained in greenhouses (22–28 °C, 70–80% relative humidity (RH), 16/8 h light/dark photoperiod with 300 μmol m^−2^ s^−1^ supplemental lighting at the canopy level), watered and fertilized appropriately, visually free from any disease or pest, and cultured for half a year before use. The orange plants had previously been lightly pruned, so that numerous flushing shoots were present at the time of testing.

Asian citrus psyllid (*D. citri*) is continuously reared at the Guangdong Engineering Research Center for Insect Behavior Regulation, SCAU, Guangzhou, China. This culture is maintained on orange seedlings at 26 ± 1 °C, 70–80% RH, and 16/8 h light/dark photoperiod. For insect preference and performance bioassay, ACP adults (7–15 days old) were used.

### 4.2. Real-Time Determination of Sulfur Volatiles from Guava Plants

Real-time measurements of volatiles were accomplished with a high sensitivity PTR-MS from Ionicon, as described in detail in [66]. The calibration of the system was performed with a standard gas mixture, as described in [67]. In this study, we focused on sulfides, including methanethiol (*m/z* 49), dimethyl sulfide (DMS, *m/z* 63), and dimethyl disulfide (DMDS, *m/z* 95). A PTR-TOF 1000 analyzer (Ionicon Analytik, Innsbruck, Austria) was used in selective ion monitoring mode, with the settings optimized for measuring sulfides [68]. The pressure of the drift tube was 2.05 mbar, and the voltage of the drift tube was adjusted so that the E/N was 117. The scanning range of time-of-flight (TOF) mass spectrometry was 30–250 amu.

The guava plant was placed into a glass cage (length × width × height: 60 × 60 × 80 cm, with a small movable door), and purified and humidified air (1 L min^−1^) was continuously maintained through the glass cage during the measurement period (Appendix A). Headspace samples were continuously determined through PTR-MS, with a flow rate of 50 mL min^−1^. For mechanical damage, the movable door of the glass cage was opened and 6 holes (0.5 cm in diameter) were quickly punched in the guava leaves, which were equally distributed in the upper, middle, and lower parts of the canopy. The measurements were started at 8:00 a.m. each day and continued for 8 h. The laboratory was kept constant at 26 ± 1 °C and 70–80% RH throughout the experiments, while a 16/8 h light/dark photoperiod was maintained with LED plant lights (5 × 24 W, Philips, Shanghai, China). The PTR-MS data were averaged over a 10 min period before and after each time point. Six replicate guava plants were used in this experiment.

### 4.3. Exposure of Orange Plant to Sulfur Volatiles

To investigate which sulfur volatiles were responsible for plant–plant communications, undamaged orange plants were independently exposed to authentic standards. The volatile dispensers (1.5 mL polyethylene flat bottom) were produced according to a previous study [14]. Methanethiol (10% in propanediol), DMS (99%), and DMDS (98%), purchased from Shanghai Macklin Biochemical Co., Ltd., Shanghai, China, were dissolved in dichloromethane to 1 μg μL^−1^, 5 μg μL^−1^, and 10 μg μL^−1^, respectively. Sulfur volatile solutions (100 μL) were pipetted into the volatile dispensers. The vial bodies were wrapped with aluminum foil for heat protection and to avoid photodegradation. The dispensers were introduced into the glass cages, as above (Appendix A). With these, we determined that the concentrations of methanethiol, DMS, and DMDS in the glass cages were 0.34, 1.21, and 2.08 ppbv (parts per billion per volume), respectively, calculated by PTR-MS (using the abovementioned method) on the second day, which were 1.78-, 1.30-, and 1.14-fold higher compared with the undamaged guava-odor source (Appendix A). For the mechanical damage group, the concentration of DMDS in the dispenser was 100 μg μL^−1^ (10 × DMDS), and the concentration in the glass cage was 45.71 ppbv, which was similar to the concentration released by the guava plants after being mechanically damaged for 15 min.

A single orange plant (with tiny buds) and a volatile dispenser (containing with sulfur volatile) were placed in the same glass chamber (Appendix A), and purified and humidified air was pumped into the system at 1 L min^−1^. The orange plants exposed to MeSA (purity 98%, Sigma-Aldrich, Shanghai, China) at a concentration of 5 μg μL^−1^ were used as the positive control. The dispensers that contained only 100 μL of pure dichloromethane were set as the solvent controls. The dispensers were replaced every 3 days. The orange plants were maintained at 26 ± 1 °C, 70–80% RH, and a 16/8 h light/dark photoperiod with LED plant lights (5 × 24 W, Philips, Shanghai, China). According to previous results regarding the induced defense effect of guava volatiles on orange plants [14], the orange plants that had been exposed for 14 d (sulfur volatiles) or 3 d (MeSA) were used for insect preference and performance bioassay, used for plant volatile collection, or stored at –80 °C for phytohormone level, phenolic content, enzyme activity, and untargeted metabolomics analyses. Furthermore, the expression of phenylalanine ammonia lyase (PAL), salicylate-O-methyl transferase (SMT), and pathogenesis-related protein 1 (PR1) genes in orange leaves that had been exposed for 3, 7, and 14 d (sulfur volatiles), or 3 d (MeSA) was also tested.

### 4.4. Insect Preference and Performance Experiments

The behavioral responses of ACP adults to sulfur volatile-exposed orange plants with olfactory and visual cues were examined in a wind tunnel, as described by [27] with some modifications. The wind tunnel was composed of a plastic glass box (150 × 40 × 60 cm) with two openable doors (40 × 60 cm) on opposite sides of the box. In the center of each door, there was a hole with a diameter of 5 cm. A treated orange plant was introduced into the box (20 cm from the openable door). Then, an electric fan was coupled on the hole and the wind speed in the box was set to 0.25 m s^−1^. On the opposite side of the box, the hole was used to release insects into the device and covered with a piece of gauze. In this way, the airflow could pass through the internal part of the box carrying the plant volatiles in the direction of the insects. A total of 30 7- to 15-day-old ACP adults (~1:1 male: female ratio) were released, and 5 replications per treatment were performed (each box was considered a replication). The number of settled ACP on orange plants was assessed at 24 h after release. The experiments were performed under 26 ± 1 °C, 70–80% RH, and 16/8 h light/dark photoperiod conditions.

The oviposition of ACP females and the subsequent performance of the next generation on sulfur volatile-exposed orange plants were recorded [69]. Briefly, sulfur volatile–exposed plants and control plants (with three tender shoots, 2–3 cm) were separately introduced into the screen cages (60 × 60 × 80 cm) for oviposition. A total of 6 10- to 15-day-old ACP adults (sex ratio = 1:1) were introduced into each screen cage and allowed to oviposit for 48 h. Then, the adults were removed, and the number of eggs were carefully recorded under a microscope. The plants with ACP eggs were continuously cultivated, and the number of nymphs and their nymphal instars were recorded at 10 d. The cages were checked for adult emergence, and new adults were recorded daily and removed. The experiments were replicated five times (each plant was considered a replication). The experiments were conducted under the same room conditions as described for the wind tunnel tests.

### 4.5. Phytohormone Quantification

Frozen orange leaf samples and small steel balls were placed in microfuge tubes and homogenized in a freezing grinder (Shanghai Jingxin, Shanghai, China) at 60 Hz for 2 min. Then, 50 mg of leaf powder from each sample was extracted with 1 mL of ice-cold 10% methanol/water (*v/v*) containing 0.125% acetic acid with constant shaking at 200 rpm (30 min, on ice), then centrifuged at 12,000 rpm for 10 min (4 °C). The supernatants from each sample were filtered through 0.22-μm microfilters, and aliquots were transferred to LC vials. The samples were analyzed using an Agilent 1200 HPLC system with an Agilent Poroshell EC-C18 column (150 mm × 3.0 mm, 2.7 μm, Agilent Technologies, Santa Clara, CA, USA) coupled to an AB Sciex Triple Quad 4000 mass spectrometer (AB Sciex, Framingham, MA, USA), as described previously [14]. Five independent plant replicates were used for each treatment.

### 4.6. Gene Expression Analysis

Total RNA was extracted from 100 mg of each leaf sample using an RNAprep Pure Plant Kit (Tiangen, China) following the manufacturer’s protocol. The first-stand cDNA was synthesized from 1 μg of total RNA per 20 μL reaction volume using a PrimeScript^TM^ RT Reagent Kit with gDNA Eraser (Takara Biotechnology, Dalian, China). Quantitative real-time PCR (qRT-PCR) analysis was performed in a CFX96 Real-time PCR (Bio-Rad, Hercules, CA, USA) with a SYBR Green Premix Pro Taq HS qPCR Kit (Accurate Biotechnology, Changsha, China) in a 20-µL reaction volume, under the following conditions: 95 °C for 30 s, 40 cycles of 95 °C for 15 s, and 60 °C for 30 s, followed by a final melting stage from 60 °C to 95 °C (0.5 °C per min increment). Specific primers are listed in Appendix A and the primer pair of PR1 refers to that previously used in [70]. Primers were synthesized by Tsingke Biotechnology Co., Ltd. (Beijing, China). The 2^−^^ΔΔCt^ method was used to calculate the relative expression levels [71], and data were normalized from FBOX and UPL7, which are widely used as the reference genes of citrus [72]. Leaves from two orange plants were combined as an individual sample, and three biological replicates were used for each treatment.

### 4.7. Determination of Enzyme Activity and Total Polyphenol Content

The activities of PAL, PPO, and POD were assayed using commercial kits purchased from Solarbio Science & Technology Co., Ltd. (Beijing, China). Briefly, leaf samples were ground to a powder in liquid nitrogen, and 100 mg of powder was homogenized and sonicated (on ice) with 1 mL kit extract solution. Homogenized samples were centrifuged (12,000 rpm, 10 min, 4 °C), and supernatants were collected. The assay of PAL, PPO, and POD was measured according to the kit protocols. The protein concentration of the enzyme extract was measured using the Bradford method [73].

The total polyphenol (TP) in the orange leaves was extracted and measured using a test kit obtained from Solarbio Science & Technology Co., Ltd. (Beijing, China). A total of 50 mg freeze dried leaf powder was mixed with 2.5 mL extract solution, and then sonicated (30 min, 60 °C) and centrifuged (12,000 rpm, 10 min, 25 °C). The supernatants were assayed according to the kit protocols. Two orange plants were combined as an individual sample, and three biological replicates of every treatment were established.

### 4.8. Analysis of Volatile Metabolites

The headspace of sulfur volatile-exposed and non-exposed orange plants was collected on HayeSep Q adsorbent using dynamic headspace sampling in a climate-controlled cabinet for 6 h [14]. The traps were extracted with 500 μL hexane containing nonyl acetate (1.75 ng μL^−1^) as the internal standard, and concentrated to 100 µL for GC-MS analysis. The samples were injected on an Agilent 8890 GC-5977B MSD equipped with a HP-5MS column (30 m × 0.25 mm × 0.25 μm, Agilent Technologies, Santa Clara, USA) with spitless mode (helium, 1.0 mL min^−1^). Column temperatures were programmed from 40 °C for 1 min, raised to 200 °C at 8 °C min^−1^, isotherm of 1 min, then raised to 280 °C at 20 °C min^−1^, and isotherm of 5 min. Injector and detector temperatures were 250 °C and 280 °C, respectively. Volatile compounds were identified according to their RIs (calculated using C7−C30) and mass spectra stored in the NIST library, and were identified by co-injection of authentic standards. The quantification of identified volatiles was conducted on an Agilent 7890B GC-FID equipped with an HP-1 column (30 m × 0.32 mm × 0.25 μm), using the same temperature programs as in the GC-MS analysis. Six independent plant replicates were used for each treatment.

### 4.9. Analysis of Semipolar Metabolites

For untargeted analysis of semi-polar metabolites, 200 mg of orange leaf powder was added to an ice-cold sample extraction solution (methanol acidified with 0.125% formic acid in vol/vol, and containing 4 ng μL^−1^ of 2-chloro-l-phenylalanine) and homogenized at 60 Hz for 90 s [74]. The homogenized samples were sonicated (30 min) and centrifuged (12,000 rpm, 10 min, 4 °C), and supernatants were passed through a 0.22-μm membrane filter. Leaves from two orange plants were combined as an individual sample, and three biological replicates were used for each treatment. The extracts were analyzed using an Ultimate 3000 UHPLC coupled with a Q Exactive mass spectrometer (Thermo Scientific, Waltham, MA, USA). At 40 °C, 2 μL of each sample was injected onto an ACQUITY UPLC^®^ HSS T3 (150 × 2.1 mm, 1.8 μm, Waters, Milford, MA, USA) with a flow rate of 250 μL min^−1^. Optimized UHPLC and mass spectrometer settings in the ESI (+) and ESI (−) modes are shown in Appendix A. Data files were processed with XCMS for feature detection, retention time correction, and alignment [75,76]. The metabolites were identified using MassBank (http://www.massbank.jp; accessed on 11 December 2021), mzClound (https://www.mzcloud.org; accessed on 11 December 2021), KEGG (http://www.genome.jp/kegg; accessed on 11 December 2021), LipidMaps (http://www.lipidmaps.org; accessed on 11 December 2021), HMDB (http://www.hmdb.ca), and a standard database (accessed on 11 December 2021) built by BioNovoGene Co., Ltd. (Suzhou, China).

### 4.10. Statistical Analysis

Data were submitted to homogeneity of variance (taking log or square root), and Tukey’s test and Student’s *t*-test were used for multiple comparisons and two groups of comparisons, respectively. Multidimensional analyses of metabolites data were performed using MetaboAnalyst [77]. Principal component analysis (PCA) and partial least squares-discriminant analysis (PLS-DA) were used to visualize the clustering among samples. Hierarchical clustering analysis (HCA) distances were calculated using Pearson correlation. A *p* value < 0.05 (Student’s *t*-test) and variable importance in the projection (VIP) >1 were used as thresholds to define the differentially expressed metabolites (DEMs). The DEMs were then mapped to the KEGG pathway for the biological interpretation of higher-level systemic functions.

## 5. Conclusions

Our results demonstrate that guava plants can continuously release sulfur volatiles, including methanethiol, DMS, and DMDS. The exposure of orange plants to DMDS resulted in the suppression of the developmental performance of ACP. The content of SA and total polyphenol, the expression of PAL, SMT, and PR1 genes, and the activities of the PAL, PPO, and POD enzymes of orange plants were increased to different degrees. Volatile and non-volatile metabolic profiles were significantly altered, and defense-related pathways were especially activated. These results indicate that the DMDS released from guava plants plays a synergistic role in the management of ACP, because DMDS can, not only repel ACP [31], but also boost anti-herbivore activities in orange plants. These results provide new insights into the roles of DMDS in the ecological management of ACP, and contribute to the understanding of the mechanism of volatile mediated plant–plant–insect interactions in the guava–citrus intercropping system. It will be important to further evaluate the field application of DMDS in citrus orchards, as well as the tradeoff between the ecological costs and the fitness benefits of DMDS-induced resistance. In addition, enhancing plant immunity will depend on the plant’s life history, physiology, and other ecological factors, to determine whether a plant would benefit from eavesdropping volatile cues and what impact volatile-mediated eavesdropping might have on plants and their insect pests [60]. An important goal for future research is to determine how citrus plants integrate volatile cues, including volatile sulfurs and terpenoids in their environment, to optimize defense investments.

## Figures and Tables

**Figure 1 ijms-23-10271-f001:**
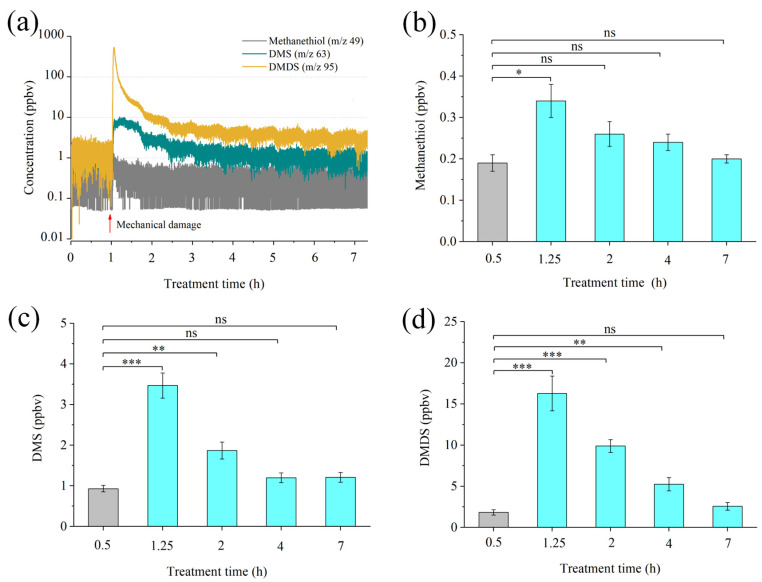
Determination of sulfur volatiles from guava plants with or without mechanical damage with a proton-transfer-reaction mass spectrometer (PTR-MS). (**a**) Real-time determination of sulfur volatiles from guava plants. (**b**–**d**) The concentration of methanethiol, dimethyl sulfide (DMS), and dimethyl disulfide (DMDS) at each time point. Data are shown as means ± SE (*n* = 6). Asterisks indicate significant differences between mechanical damaged and control (undamaged) plants based on Student’s *t*-test (* *p* < 0.05, ** *p* < 0.01, and *** *p* < 0.001); ns, no significant difference.

**Figure 2 ijms-23-10271-f002:**
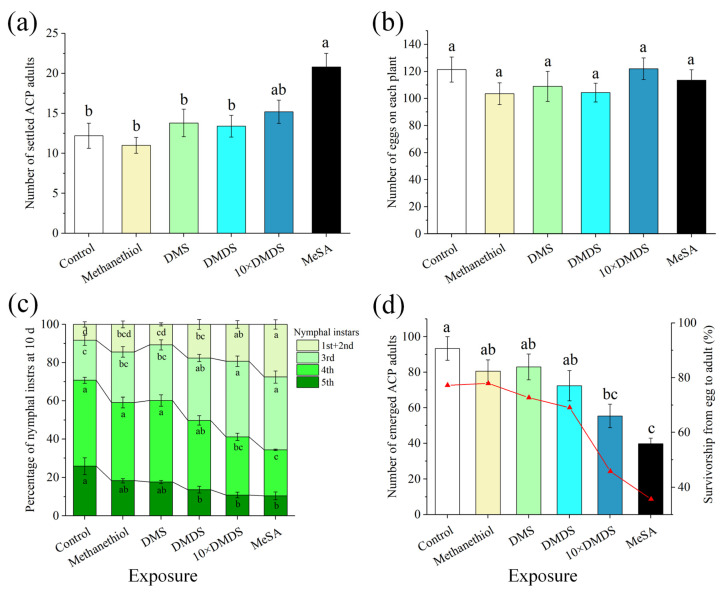
Exposure of sweet orange plants to sulfur volatiles affected the host selection behavior and population development of Asian citrus psyllid (ACP). (**a**) The number of settled ACP adults at 24 h after their release in wind tunnels. (**b**) The number of eggs on each plant after ACP oviposition for 48 h. (**c**) The percentage of nymphal instars at 10 d after ACP release for oviposition. (**d**) The number of total emerged adults (columns) and the rate of survivorship (dotted line) from egg to adult. Data are shown as means ± SE (*n* = 5). Different letters on the columns indicate significant differences among the different treatments based on Tukey’s test (*p* < 0.05).

**Figure 3 ijms-23-10271-f003:**
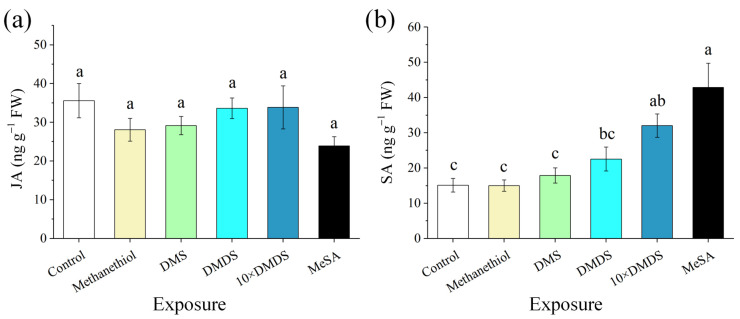
The levels of jasmonic acid (JA) and salicylic acid (SA) in orange plants after exposure to sulfur volatiles. (**a**) JA content; (**b**) SA content. Data are shown as means ± SE (*n* = 5). Different letters on the columns indicate significant differences among different treatments based on Tukey’s test (*p* < 0.05).

**Figure 4 ijms-23-10271-f004:**
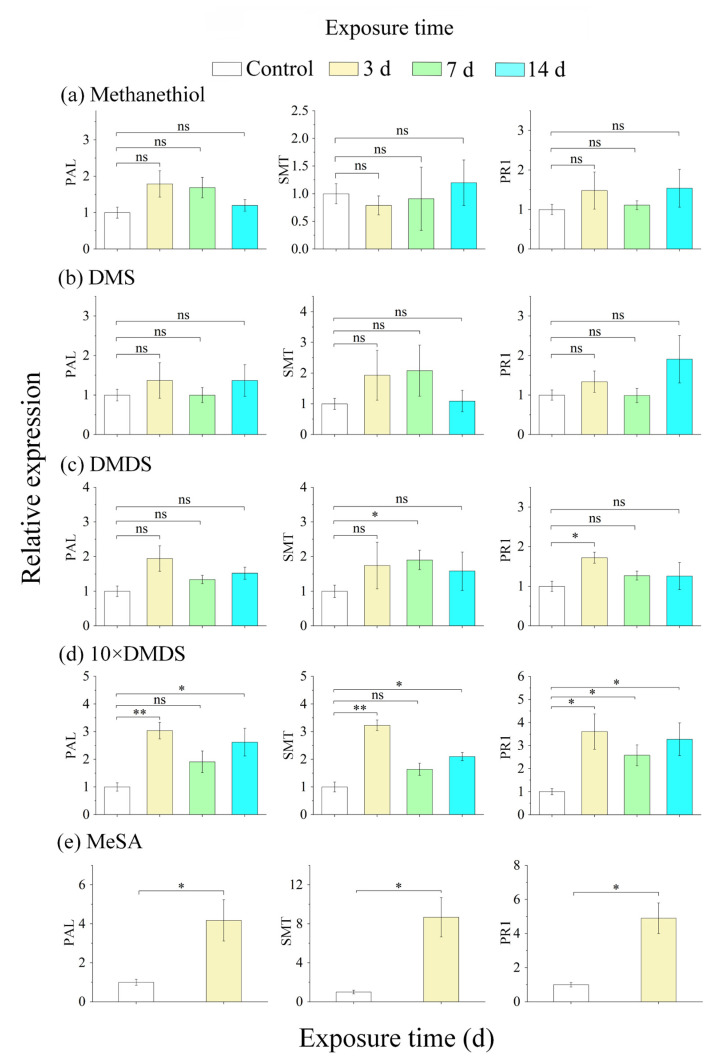
The expression of phenylalanine ammonia lyase (PAL), salicylate-O-methyl transferase (SMT), and pathogenesis-related (PR1) genes of orange plants after exposure to sulfur volatiles. Data are shown as means ± SE (*n* = 3). Asterisks indicate significant differences between exposed and control plants, based on Student’s *t*-test (* *p* < 0.05 and ** *p* < 0.01); ns, no significant difference.

**Figure 5 ijms-23-10271-f005:**
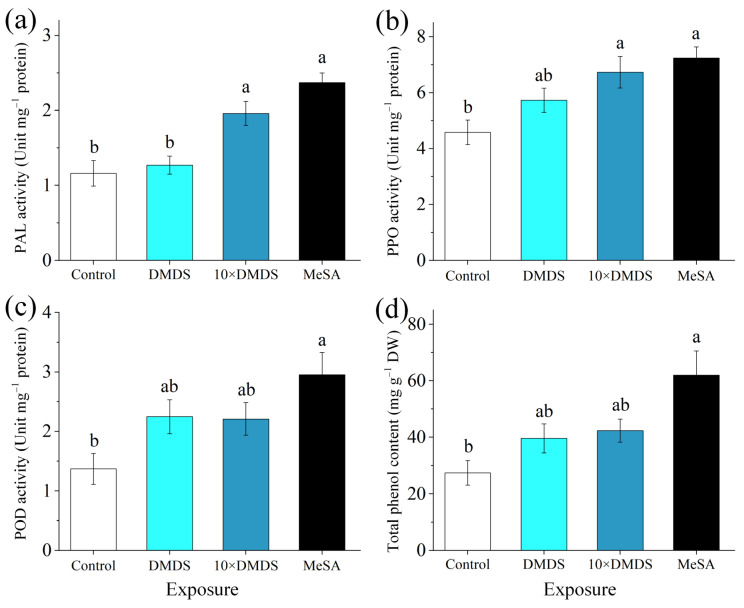
Phenylalanine ammonia lyase (PAL), polyphenol oxidase (PPO), and peroxidase (POD) activities and total phenol content of orange plants after exposure to dimethyl disulfide (DMDS). (**a**–**c**) PAL, PPO, and POD activities, respectively; (**d**) total phenol content. Data are shown as means ± SE (*n* = 3). Different letters on the columns indicate significant differences among different treatments based on Tukey’s test (*p* < 0.05).

**Figure 6 ijms-23-10271-f006:**
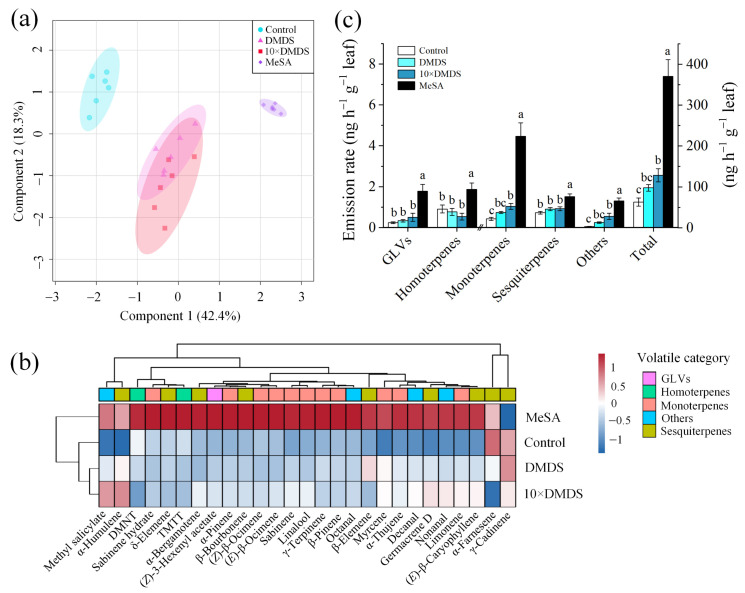
Volatile profiling of orange plants after exposure to dimethyl disulfide (DMDS). (**a**) PLS-DA (score plot); and (**b**) clustering analysis and heat map (average) of orange volatile metabolites. (**c**) Emission of orange volatiles from different classifications. Data are shown as means ± SE (*n* = 6). Different letters on the columns indicate significant differences among different treatments based on Tukey’s test (*p* < 0.05).

**Figure 7 ijms-23-10271-f007:**
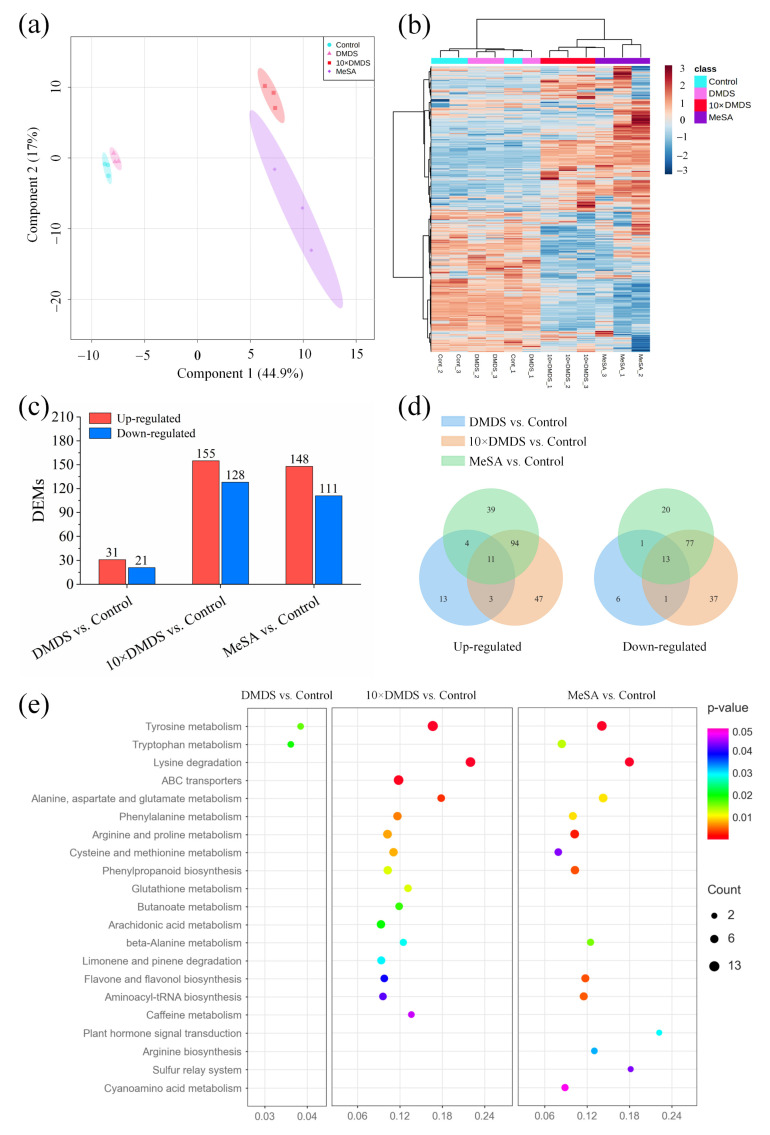
Characterization of the differentially expressed semipolar metabolites (DEMs) of orange plants after exposure to dimethyl disulfide (DMDS). (**a**) PLS-DA (score plot); and (**b**) clustering analysis and heat map of expression measures of semipolar metabolites. (**c**) The number of DEMs in different groups (for details, see Appendix A). (**d**) Venn diagram showing the distribution and overlap of up- and downregulated DEMs across all treatments. (**e**) KEGG pathway enrichment of DEMs.

## Data Availability

Not applicable.

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
