# Peer review of "Volatile Dimethyl Disulfide from Guava Plants Regulate Developmental Performance of Asian Citrus Psyllid through Activation of Defense Responses in Neighboring Orange Plants"

_ijms, 2022, doi:10.3390/ijms231810271_

Round 1
Reviewer 1 Report
Dear authors,
Manuscript ijms-1865907 entiteled "Volatile dimethyl disulfide from guava plants can regulate developmental performance of Asian citrus psyllid through activation of defense responses in neighboring orange plants" and authored by Siquan Ling , Hualong Qiu , Jinzhu Xu , Yanping Gu , Jinxin Yu , Wei Wang , Jiali Liu , Xinnian Zeng targets a hot topic and is potentially very interesting to the journal readers and the sccientific community as a whole. While research design is appropriate and even sound, quality of presentation of data is vey weak and needs deep revision and improvement:
1. English of the manuscript is very weak and I can never suggest your manuscript to publication with the current level of english. The manuscript have to be addressed with a native english speaker or submitted to journal service improvement.
2. Abstract section : Please highlight at the end of the abstract the practical implication of your work. What your study could add to the field ! a nice issue is to relate your findings to the tradeoff that you pointed out between volatile induced defense changes and resistance to biotic stresses.
3. Introduction: please dedicate a part of your introduction to volatiles of guyava plants. This part is very important to the readers ! Please present what have been done in this field before your study.
4. Discussion : Please discuss the cost of the strategy you develop in your paper and the tradeoff between volatile induced defense changes and resistance to biotic stresses.
5. Please dedicate a section to conclusion. It is a pitty that such nice findings could not be highlighted with practical implications to end users and farnmers. Please highlight also ootlook with futrure directions in the field of research.
6. References : please add relevant references to the section in introduction dedicated to volatiles from Guava plants.
I am really happy to receive an improved version of the manuscript that I could recommend for publication and that meets the journal standards.
Best regards
Author Response
Point 1: English of the manuscript is very weak and I can never suggest your manuscript to publication with the current level of English. The manuscript has to be addressed with a native English speaker or submitted to journal service improvement.
Response 1: Thank you very much for your kind review. We apologize for the poor English of our manuscript. The manuscript undergone English language editing by MDPI, which uses experienced, native English-speaking editors. The text has been checked for correct use of grammar and common technical terms. We really hope that the English writing has been substantially improved. These changes will not influence the content and framework of the manuscript. The line numbers for answer in our responses are referred to the revised manuscript. And here we did not list the changes but marked in red in the revised manuscript.
Point 2: Abstract section: Please highlight at the end of the abstract the practical implication of your work. What your study could add to the field! a nice issue is to relate your findings to the tradeoff that you pointed out between volatile induced defense changes and resistance to biotic stresses.
Response 2: We thank the reviewer for pointing out this issue. We have highlighted a sentence at the end of the abstract, “Our results indicated that DMDS from guava plants can activate defense responses in eavesdropping orange plants and boost their herbivore resistance to ACP, which suggests the possibility of using DMDS as a novel approach for the management of ACP in citrus orchards.” (L30-32 of the revised manuscript)
Point 3: Introduction: please dedicate a part of your introduction to volatiles of guava plants. This part is very important to the readers! Please present what have been done in this field before your study.
Response 3: We agree. Guava volatiles have attracted intensive interest recent years. We briefly described the research on guava volatiles, especially terpenes and sulfur volatiles. For details, see L60-81 for details.
Point 4: Discussion: Please discuss the cost of the strategy you develop in your paper and the tradeoff between volatile induced defense changes and resistance to biotic stresses.
Response 4: The comments are valuable and helpful for revising and improving our manuscript and have provided good guidance for our studies. It is known that plant defense is costly as the activation of defense responses requires a significant investment of resources toward defense production. Thus, it will be important to evaluate the ecological costs and fitness benefits of DMDS-induced defense responses in citrus plants in the future studies. We have discussed the tradeoff between costs (induced defenses) and benefits (resistance to insect pest) in the Discussion (L338-357).
Point 5: Please dedicate a section to conclusion. It is a pity that such nice findings could not be highlighted with practical implications to end users and farmers. Please highlight also outlook with future directions in the field of research.
Response 5: We agree. In fact, we have summarized the highlights in the last paragraph of the Discussion section in the first submitted manuscript. According to the reviewer’s comments, we have rewritten the Conclusion section and refined the conclusions in the revised manuscript (L546-565).
Point 6: References: please add relevant references to the section in introduction dedicated to volatiles from Guava plants.
Response 6: The relevant references have been added in the Introduction section according to the comments. We tried our best to improve the manuscript. We will be happy to edit the text further, based on helpful comments from the reviewers.

Reviewer 2 Report
The authors of this manuscript present an interesting research study regarding the volatile dimethyl disulfide from guava plants that can regulate developmental performance of Asian citrus psyllid through activation of defense responses in neighboring orange plants. Introduction and material and methods are well described. Despite results presented in tables and figures are clear and quite explicable please check the style of them. Due to limited research studies, I believe that the authors discuss and explain sufficiently the findings of their work. The text needs revisions. Research studies in guava plants are not that well described in the past and thus I believe that this research study can add further interest especially in the tropical regions worldwide.
Abstract
COMMENT:
Abstract describes sufficient the findings of this research work.
Introduction
Introduction section is well written and, in my opinion, give the appropriate information without being extended. The purpose of the research work is clearly presented.
Please check style of references [1,2] instead of [1, 2]. Please check the whole document.
Results
According to my opinion I think that figures and tables must be included in the main text and not at the end of the manuscript. Please advise author’s instructions.
Please check and correct the style of the figures to be in accordance to the author’s instructions
Discussion
Rouseff and co-authors [28] Rouseff et al., please check the rest of the text
(10×DMDS). Volatiles please check if there is a gap between words.
Martini et al. [47] found Martini et al., please check author’s instructions about the style and correct the rest of the text
Materials and Methods
Conclusions
Please check author’s instruction whether a conclusion section is needed or not.
References
COMMENT:
I suggest to check reference list style again. Please check it according to author’s instructions.
In spite of the manuscript is clear and carefully written some improvements should be done.
Please delete the contains after 25th page
Author Response
Point 1: Abstract: Abstract describes sufficient the findings of this research work.
Response 1: We appreciate the reviewer for the kind review. We have revised the manuscript in accordance with the recommendations.
Point 2: Introduction: Introduction section is well written and, in my opinion, give the appropriate information without being extended. The purpose of the research work is clearly presented. Please check style of references [1,2] instead of [1, 2]. Please check the whole document.
Response 2: Thank you for your kind comments. The reference style has been checked and revised according to the author’s instructions.
Point 3: Results: According to my opinion I think that figures and tables must be included in the main text and not at the end of the manuscript. Please advise author’s instructions. Please check and correct the style of the figures to be in accordance to the author’s instructions
Response 3: The comments are valuable and helpful for revising and improving our paper and have provided good guidance for our studies. According to the author’s instructions, the figures have been included in the main text in the revised manuscript.
Point 4: Discussion: Rouseff and co-authors [28] Rouseff et al., please check the rest of the text. Martini et al. [47] found Martini et al., please check author’s instructions about the style and correct the rest of the text. (10×DMDS). Volatiles please check if there is a gap between words.
Response 4: We thank the reviewer for pointing out this issue. We have checked the whole document and revised according to the comments and author’s instructions.
Point 5: Conclusions: Please check author’s instruction whether a conclusion section is needed or not.
Response 5: Conclusion section is optional. In fact, we summarized the highlights in the last paragraph of the Discussion section in the first submitted manuscript. According to the reviewer’s comments, we have rewritten the conclusions and dedicated a conclusion section in the manuscript.
Point 6: References: I suggest to check reference list style again. Please check it according to author’s instructions. In spite of the manuscript is clear and carefully written some improvements should be done. Please delete the contains after 25th page.
Response 6: We have checked and revised the document based on the helpful comments and author’s instructions. We will be happy to edit the text further according to the helpful comments from the reviewers.

Round 2
Reviewer 1 Report
Dear authors,
I can now recommend your manuscript for publication.
Best regards